# Development, Content Validity and Usability of a Self-Assessment Instrument for the Lifestyle of Breast Cancer Survivors in Brazil

**DOI:** 10.3390/nu16213707

**Published:** 2024-10-30

**Authors:** Jaqueline Schroeder de Souza, Nathalie Kliemann, Francilene Gracieli Kunradi Vieira, Aline Al Nahas, Luiza Kuhnen Reitz, Elom Kouassivi Aglago, Cândice Laís Knöner Copetti, Lilian Cardoso Vieira, Inge Huybrechts, Nivaldo Barroso de Pinho, Patricia Faria Di Pietro

**Affiliations:** 1Post Graduate Program in Nutrition, Federal University of Santa Catarina, Florianopolis 88040-900, Brazil; jaqueline.schroeder04@gmail.com (J.S.d.S.); francilene.vieira@ufsc.br (F.G.K.V.); candice.lk@hotmail.com (C.L.K.C.); livnutrisport@gmail.com (L.C.V.); 2Oncology Research Center, Florianopolis 88015-130, Brazil; nathalie.kliemann@gmail.com; 3International Agency for Research on Cancer, 69366 Lyon, France; alnahasa@iarc.who.int (A.A.N.); huybrechtsi@iarc.who.int (I.H.); 4Florianopolis Specialized Oncology Center, Florianopolis 88032-005, Brazil; luizakreitz@gmail.com; 5School of Public Health, Imperial College London, St Mary Campus, Norfolk Place, London W12 0BZ, UK; aglagoelom@gmail.com; 6Brazilian Society of Oncological Nutrition, Rio de Janeiro 20530-050, Brazil; suporte@sbno.com.br

**Keywords:** breast cancer, lifestyle, diet, physical activity, WCRF/AICR, content validity, convergent validity, usability testing

## Abstract

Background/Objectives: Breast cancer is the most common cancer among women globally, and it negatively impacts diet and quality of life, increasing the risk of recurrence. Adhering to World Cancer Research Fund (WCRF) and American Institute for Cancer Research (AICR) lifestyle guidelines, such as healthy eating habits and nutritional status, can help in primary and secondary cancer prevention. However, no questionnaire was found for self-assessment of these guidelines for the Brazilian population. The aim of this study is to carry out content validity, pilot, and usability testing of the self-administered digital instrument “PrevCancer” assessing adherence to the WCRF/AICR recommendations in Brazilian female breast cancer survivors. Methods: We conducted a psychometric study that involved the development of an instrument based on WCRF/AICR recommendations. Assessment of content validity involved the Content Validity Index (CVI) based on expert assessments (n = 7). The pilot study involved the System Usability Scale (SUS) after applying the developed instrument (n = 65) and anthropometric assessment for convergent validity by female participants (n = 55). The final usability test consisted of evaluating the satisfaction with the instrument of women with breast cancer (n = 14). Results: The “PrevCancer” instrument demonstrated good content (CVI = 1.0) as well as good usability and acceptability in the pilot study (mean SUS score = 88.1). The convergent validity stage demonstrated positive associations between the PrevCancer parameters and anthropometric parameters (*p* < 0.001). In the final usability study (mean SUS score = 90.3), participants’ receptivity to the instrument was excellent. Conclusions: The PrevCancer instrument had valid content and great usability by the target population, proving to be a useful tool for future cancer research.

## 1. Introduction

Breast cancer is the most prevalent type of cancer and the main cause of death from cancer among women in Brazil and worldwide [1,2]. In Brazil, it was estimated that between 2023 and 2025, there will be 74,000 new cases of breast cancer, representing 20% of cases among women [3,4].

After breast cancer treatment, patients may be susceptible to locoregional recurrence or metastasis [5]. Therefore, reducing exposure to known modifiable risk factors for cancer, especially those related to lifestyle, can constitute a relevant public health approach to help prevent the recurrence of the tumor [6,7,8]. It is well understood that nutrition plays a key role among protective factors against cancer development and recurrence. A diet rich in plant-based foods, such as fruits, vegetables, whole grains, and legumes, contains important micronutrients and phytochemicals that act as antioxidants and have anti-inflammatory and anti-cancer properties. On the other hand, high consumption of ultra-processed foods, rich in sugar, saturated fats, and salt, as well as high alcohol consumption, can increase the production of toxic metabolites, which promotes oxidative stress and increase risk of cancer [9,10]. In addition, these highly processed foods and alcoholic beverages promote weight gain, which interferes with the secretion of inflammatory mediators and metabolic and hormonal dysregulation, thus being proven to be related to carcinogenesis [6,7,8].

Based on analyses of the evidence on how diet and other lifestyle factors are associated with the risk of developing cancer, the World Cancer Research Fund (WCRF) and American Institute for Cancer Research (AICR) released, in 2018, a set of evidence-based recommendations for cancer prevention and survival [7]. These lifestyle guidelines are the best known globally and are intended for primary and secondary cancer prevention [11]. Adherence to these recommendations has been associated with reduced risk of different types of cancer [12,13], including breast cancer [14], with 25% of breast cancer cases occurring among women with low and moderate adherence to the WCRF/AICR recommendations. In the context of recurrence, it has already been identified that patients with higher adherence scores to the WRCF/AICR recommendations had a lower risk of bladder cancer recurrence [15], and it is known that healthier lifestyle habits have an impact in lower chance of breast cancer [16], which is the focus of this work. According to Mirizzi et al. [17], adherence to WCRF/AICR recommendations has also been inversely associated with all-cause mortality. These findings reinforce the importance of following these guidelines for the prevention of cancer and recurrence.

In the Brazilian context, there are some peculiarities of lifestyle habits, such as the consumption of beans, which typically make up Brazilian dishes, and the consumption of chimarrão, very common in the southern region of the country [18]. Chimarrão is an aqueous infusion made from the dried leaves of the *Ilex paraguariensis* plant, normally consumed at very high temperatures, after adding almost boiling water to the herb. The consumption of chimarrão at high temperatures impacts the country’s highest incidence rates of esophageal cancer in the southern states. Therefore, correct guidance regarding the consumption of chimarrão at a warmer temperature is necessary. It is noteworthy that, even for patients who had a primary diagnosis of breast cancer, there is a considerable risk of developing esophagus cancer as a secondary cancer [19,20], which makes the inclusion of the chimarrão consumption component relevant in the context of Brazilian reality. Based on these specificities, in 2020, the National Cancer Institute adapted the WCRF/AICR recommendations [7] to the context of life in Brazil, launching a report with guidelines for the national reality [18].

In order to better put into practice the adherence to WCRF/AICR recommendations, a score was developed for scientific research [21]. In addition, individual self-assessment digital questionnaires were developed in England (the “Cancer Health Check” [22]) and in France (“Prévention cancers: le test” [23]) to facilitate self-assessment on adherence to WCRF/AICR recommendations. However, no online questionnaire was found allowing self-assessment on adherence to the WCRF/AICR recommendations [7] for the Brazilian population. Lifestyle self-assessment is relevant because being aware of current behaviours and score of the WCRF/AICR recommendations may increase motivation towards behavioral changes and individuals may become more prone to receive advice from health professionals regarding their lifestyle habits [24,25]. Previous studies have shown that lifestyle self-assessment instruments may help identify risk factors and may positively influence health status, preventing cancer recurrence and reducing healthcare costs [26,27,28,29]. Thus, self-assessment digital questionnaires focused on lifestyle behaviours can be useful instruments to target individualized prevention strategies among breast cancer survivors and the general population [30,31].

Considering that there is no known digital version of the WCRF/AICR score [7,21] that can be self-administered by the Brazilian population, especially among breast cancer survivors, who stand out as a population that has worse quality of diet and life than individuals without the disease [32,33], the objective of this study was to carry out content validity, pilot testing, and usability testing of an original self-administered digital instrument to assess adherence to the WCRF/AICR recommendations in Brazilian female breast cancer survivors.

## 2. Materials and Methods

### 2.1. Research Design

This is a psychometric, descriptive, and cross-sectional study. After defining the instrument development, which consists of a methodological stage, this study consisted of four phases: (1) content validity [34]; (2) pilot study, which included usability and acceptability tests and the convergent validity test of the anthropometric measures; (3) instrument adaptation; (4) final usability study (Figure 1).

### 2.2. Instrument Development

The first version of the instrument was developed in Brazilian Portuguese. A bibliographical survey was carried out on existing instruments [22,23,26,27,28,29] regarding lifestyle screening of patients with a previous diagnosis of breast cancer. Additional research was carried out regarding national population food consumption surveys [35,36] to identify the foods most consumed by Brazilians and to provide more appropriate examples of foods for the instrument’s questions. The Canva graphic design platform was used to obtain images to help clarify the content of the questions.

The content and scoring format of the digital self-assessment questionnaire was mainly based on the WCRF/AICR report [7], the summary of the third expert report with a Brazilian perspective [18], and the instrument proposed by Shams-White et al. [21], which assesses the score for eight WCRF/AICR recommendations.

The first part of the questionnaire comprised the WCRF/AICR [7] recommendations, which were transformed into easy-to-understand questions about lifestyle habits aimed at breast cancer survivors, grouped in the following topics: date of birth (to enable age calculation); current body weight and height; perception of body image; practice of moderate and vigorous physical activity; number of portions of fruits and vegetables consumed; frequency of consumption of beans, sugary drinks, ultra-processed foods, whole grains, nuts and seeds, red meat, and processed meat; number of doses of alcohol consumed per week; total duration of exclusive breastfeeding; chimarrão consumption (matte drink); smoking and sun exposure, as described below (Appendix A):

Weight and height component: the system automatically calculates the Body Mass Index (BMI) according to reported body weight and height data and classifies it according to age groups, those being 18 to 59 years old according to the World Health Organization [37,38] and 60 years of age or older according to Pan American Health Organization [39].

Body image component: it was decided to replace the waist circumference parameter, which is mentioned by WCRF/AICR [7] as one of the ways to monitor nutritional status, with the perception of body image as evaluated through the Stunkard Silhouette Scale (SSS) [40]. This replacement was motivated considering the following aspects: (1) most individuals are unaware of their measurement of personal waist circumference, which makes self-reporting of this parameter difficult [41]; (2) there are different techniques for measuring the waist [42,43,44] and, although the measurement was standardized for the present study, there could be confusion on the part of participants regarding the correct anatomical point for self-measurement; (3) it has already been demonstrated in studies that the SSS has a high predictive capacity for identifying overweight/obese individuals [45,46] and is associated with the measurement of waist specifications [47].

Physical activity component: the “physical activity” component was based on recommendations established in the literature [7,18]. Questions regarding this subject were developed based on the Cancer Health Check instrument [22]. Various examples of physical exercise modalities were included, so that the participant can recognize the intensity category of the specific physical exercise(s) they practice [48]. Based on the weekly time (in minutes) of physical activity practiced, the system automatically adds up the minutes related to moderate and vigorous/intense physical exercise, making automatic classification possible based on the physical activity cut-off points defined by Shams-White et al. [21].

Fruits and non-starchy vegetables component: the system made it possible to automatically calculate the total weight of fruits and vegetables consumed by asking information on the number of portions consumed of these foods (80 g per serving on average) [18] and by providing examples of one portion of the foods most commonly consumed according to the 2017–2018 Brazilian Family Budget Survey Report [36].

Beans component: the instrument included a question related to the consumption of beans, including legumes of all colors, lentils, soybeans, chickpeas, and peas [49,50]. Based on the indication of the weekly frequency of consumption, the system automatically calculated the score according to the recommendations of the Food Guide for the Brazilian Population [49].

Sugary drinks component: consumption of sugary drinks was assessed according to the weekly frequency of consumption. We provided examples based on the international version of the instrument [22], such as soft drinks, ready-to-drink industrialized juices with sugar, natural juices (with or without added sugar), ready-to-drink industrialized teas (such as bottled or canned industrialized mate tea), coffee and tea with added sugar, and energy/sports drinks [7,18,22].

Ultra-processed foods component: to evaluate the consumption of ultra-processed foods, the NOVA classification [51] was used as reference, with some foods being exemplified in the structure of the question: stuffed biscuits, packaged snacks, instant noodles, ice cream, chocolates, breakfast cereals, and “fast food” type foods, such as pizzas and French fries [7,18,49,50].

Whole grains, nuts, and seeds component: a question was defined about the consumption of whole grains, nuts and seeds, which contribute to the daily fiber intake [7,18,50]. The answer options are related to the daily frequency of consumption [22].

Red meat and processed meat component: red meat consumption also followed the NOVA classification system [51] for the instrument and included examples of red meat such as beef steak, ground beef, lamb meat, and pork. The alternative answers involved weekly frequency of consumption [22]. The highest score in this question was for the answer “3 servings or less per week”, as this is equivalent to 350 to 500 g of prepared meat [7,18]. One of the component’s questions, in turn, focused on processed meat and included the consumption of foods such as industrialized or “fast food” hamburgers, bacon, sausages, sausages, salami, ham, smoked turkey breast, mortadella, and other processed meats [51]. The score was based on a previously published assessment score for adherence to WCRF/AICR recommendations [21].

Alcoholic drinks component: a question on weekly consumption of alcoholic beverages was developed, which include wine, sparkling wine/champagne, beer, draft beer, whiskey-based distilled drinks, vodka, rum, cachaça (common Brazilian rum), among others [7,18,51]. The answer options included specific fields to fill in the number of portions consumed according to the type of alcoholic beverage. Examples of beverage portions in volume (mL) and type of container (cup, can, or dose) were added in accordance with the previously consulted bibliography [52,53]. The system automatically converted the portions of alcoholic beverages consumed per week into grams of ethanol consumed per day. For this function, the alcoholic levels of each drink established in the literature were considered [54,55].

Breastfeeding component: this component involved a question regarding the sum of the months of exclusive breastfeeding (without other foods and liquids in the child’s diet) of all the participant’s children. Examples of how to add up the total time of exclusive breastfeeding were provided. The question had a score similar to what was previously recommended [21].

Chimarrão (matte drink) component: the “chimarrão consumption” (matte drink) component was included, considering that it is part of one of the recommendations added in the Portuguese version of the WCRF/AICR report [18]. This parameter was structured based on two factors: whether the participant consumes this type of drink and, if the participant consumes, what is the average temperature at which the chimarrão is usually consumed by the participant. The reference for very hot chimarrão is the formation of bubbles while heating the water [18].

Smoking habits component: participants were asked about smoking habits based on existing screening questionnaires, with the following answer options: current smoker, ex-smoker, or never smoked [22,56].

Sun exposure component: the answer options for the sun exposure component were based on the Sun Exposure and Protection Index (SEPI) questionnaire, validated and translated into Portuguese [57]. The WCRF/AICR [7] have already warned about the risk of inadequate sun exposure and, therefore, it becomes relevant to evaluate direct exposure of arms, legs, and/or back to the sun and the use of sunscreen [57].

The second part of the questionnaire (Appendix A) involved sociodemographic questions for sample characterization and to provide a complete database for future research. It should be noted that this second part of the instrument is not intended to be maintained for the future official version of the instrument to be disseminated to the population; this section was only included for the purposes of descriptive analyses related to this research. This part of the questionnaire includes the following information: full name; email; telephone; city of origin; ethnicity; education; occupation; average monthly family income; and marital status. Questions related to ethnicity and average monthly family income were formulated according to the definitions by the Brazilian Institute of Geography and Statistics [58,59].

#### 2.2.1. Final Score

For the final score, each response automatically generates a specific score, which can vary from zero to one point, with the following scoring for each question: 0 points (low adherence to the recommendation), 0.5 points (moderate adherence), and 1 point (high adherence) (Appendix A). Similar to what was established by Shams-White et al. [21], the component as a whole should have a maximum score of 1 point and could be subdivided into two topics with a maximum score of 0.5 points each. For example, the red and processed meat component was divided into two questions, one for red meat and the other for processed meat. In this case, for each component question, the score was defined as 0 points (does not meet the recommendation), 0.25 points (partially meets), or 0.5 points (meets the recommendation). The total sum of the scores for all components can generate a score with a range of 0 to 10 points. The total score is proportional to compliance with WCRF/AICR recommendations for cancer recurrence prevention [7] (Appendix A).

#### 2.2.2. Automatic Feedback

Automatic feedback (Appendix A) to participant regarding the score and current lifestyle behaviours was organized as follows: (i) the respondent receives information on whether their adherence to lifestyle recommendations is low, moderate, or adequate based on the total score generated in the system; (ii) all recurrence prevention recommendations appear on screen with objective explanations of how to achieve each healthy lifestyle behavior based on the recommendations of WCRF/AICR [7], in addition to the Brazilian perspective of these guidelines [18]. Feedback to participants consisted of a PDF file containing all this data, which can be downloaded immediately after completing the questionnaire. In order to make the explanation of the total score clearer for the participant, for the automatic feedback, the total score was divided into 0 to 3 points (unhealthy lifestyle), 4 to 7 points (moderately healthy lifestyle), and 8 to 10 points (healthy lifestyle). This classification was defined based on previous studies that used alternative models of adherence scores to the WCRF/AICR recommendations [60,61,62].

#### 2.2.3. Digital Structuring

The first digital structuring of the instrument was carried out on the Google Forms^®^ (v. 2023) platform with the assistance of a researcher in the Information Systems area. Regarding compliance with General Data Protection Law, according to Dove et al. [63], this was a step designed to ensure harmony with regulations covering data collection and processing. In the digital era, following the guidelines of this law strengthens compliance with participant rights [63]. In this regard, the instrument was adapted to the standards by the company CEJUR-FGV [64]. Throughout this assessment, areas of potential risk were identified and strategies to mitigate the risks were established by the technical team together with the research team. Adapting to the General Data Protection Law involved analyzing the risk map, structuring the privacy policy, defining the terms and conditions of use, and defining the data use consent form.

### 2.3. Phase 1: Content Validity

To verify whether the content of the developed questionnaire was adequate and comprehensive, we carried out a content validity following the steps proposed by Yusoff [34]: (a) prepare the content validity form; (b) select a review panel of experts; (c) request a review of domains and items; (d) provide scores for each item; (e) carry out content validity through online discussions; (f) calculate the Content Validity Index (CVI). More details about the CVI calculation are provided in the Section 2.8.

The content validity form [34] included the following items: (i) domain; which consisted of the recommendation of the instrument component; (ii) items; which were questions about the specific recommendation; (iii) degree of relevance; which varied from 1 “totally disagree” (the item is not relevant) to 4 “totally agree” (the item is highly relevant); and (iv) field for observations. This form was sent by email to each expert. The content validity committee followed what was proposed by Lynn [65] regarding the number of members in the group, with a minimum of three and a maximum of ten experts. The selection of the panel of experts was based on the dissemination of the research topic in co-participating institutions and invitations to professionals with experience in oncology and who were interested in the study. In this way, experts were identified, and the committee was composed of researchers working in the areas of nutrition, nursing, and psychology. All experts signed the Informed Consent Form.

Based on the responses received in the content validity form from each expert, all data on degree of relevance and observations for each item and instrument domain were integrated into a specific Microsoft Excel^®^ (v. 2021) spreadsheet. Thus, all items were evaluated for the CVI tool score, which will be further detailed in the data analysis. Comments with similar and divergent opinions on the instrument’s items were analyzed, which served as an agenda for the meetings that were scheduled with the panel of experts.

Two online meetings were scheduled through the Web Conference^®^ platform to discuss the experts’ evaluations and establish consensus on divergent comments placed on the content validity form. The experts’ perceptions regarding the structuring of each component of the instrument were captured to qualify its content. Adjustments were made whenever necessary according with what was discussed with the experts during the meetings. Regarding differing opinions among experts, votes were held during the meetings on which solution to take to adjust specific items. The necessary adjustments were based on the suggestions voted by most experts, who received the final adjusted version of the instrument via email.

### 2.4. Phase 2: Pilot Study

The first version of the digital instrument structured on the Google Forms^®^ (v. 2023) was evaluated by a convenient female sample attending a local teaching and extension event in Florianopolis (Santa Catarina, Brazil), which was open to the community. Based on prior research, we followed the recommendation to recruit at least 30 participants for the pilot study, since this is considered an adequate sample for usability and convergent validity studies [66].

Participants were instructed to answer the instrument on a laptop that was available at the event or on their personal smartphone. The researchers were available to clarify any questions the participant had, taking care to make as few interventions as possible, as recommended by Da Costa et al. [67].

After completing the digital questionnaire, each participant received a brief form from the research team with questions about personal information, access to a computer/notebook at home, and their perception of the instrument [67,68]. The usability and acceptability questions were collected by the main researcher and the data was tabulated in a Microsoft Excel^®^ (v. 2021) spreadsheet. Details on how usability was assessed are described in the Section 2.8.

Participants also had their body weight, height, and waist circumference measured to assess the convergent validity [69], aiming to assess how closely the instrument’s self-reported anthropometric measures were related to objectively measured anthropometrics data. Body weight and height measurements were carried out on a mechanical anthropometric scale with a stadiometer coupled to the same equipment (Filizola^®^ brand), with a capacity of 150 kg and precision of 100 g, according to established protocols [37,70]. After measuring these parameters, the Body Mass Index (BMI) was calculated by the research team. Waist circumference was measured using an inelastic anthropometric tape (accuracy of 0.1 cm) according to the procedures and classification previously described [38].

### 2.5. Phase 3: Instrument Adaptation

All suggestions for improvements to the instrument from the pilot study were tabulated in a Microsoft Excel^®^ (v. 2021) spreadsheet and considered during this instrument’s adaptation phase. The second digital structuring of the instrument was conducted with the support of a junior Computer Science and Information Systems company [71]. Online meetings were held on Google Meet between the research team and the company’s technical team to outline the digital solution, design definitions, visual elements, and resources to be used. All this information was put together into an interface design prototype to suggest the structure of the website. To achieve this end, a virtual form developed with the “React” Javascript [72] library was defined, being able to use frameworks that are made up of it, as well as the “NEXT.JS” framework [73].

### 2.6. Phase 4: Final Usability Study

The final usability study was carried out with women according to the following inclusion criteria: diagnosis of primary breast cancer (stage 0 to IIIb); completion of adjuvant treatment (chemotherapy and/or radiotherapy) at least 6 months prior to the date of data collection; age over 18; basic skills for using a computer and cell phone; having a computer or smartphone with internet access; knowing how to read and write. The exclusion criteria considered were as follows: presence of metastasis or recurrence; pregnancy and breastfeeding; presence of any cognitive or psychiatric difficulties that make it difficult to understand the work and collect the information necessary for the research; presence of symptoms of significant nutritional impact and/or need for enteral or parenteral nutritional support; amputation of limbs. The final usability testing was carried out with at least 11 participants, based on previous recommendations from the literature [74].

Participants were recruited from three different health institutions, all of which are considered oncology reference centers in the State of Santa Catarina (Brazil). For sample characterization purposes, the research team collected sociodemographic and clinical information related to breast cancer from the institutions’ medical records, such as menopause status, stage of breast cancer at diagnosis [75], lymph node involvement, and clinical subtype of the disease.

The usability test rooms were equipped with computers and adequate furniture, free from objects that could cause distractions [76,77]. Additionally, the room doors were closed during the application of the digital instrument to avoid possible external distractions. The participant performed the test on a computer equipped with a webcam, the test moderator (main researcher) stood behind and diagonally from the participant, and the observers remained behind the moderator taking notes about the participant’s test [77]. The Hotjar^®^ software (v. 2024) remained active on the computers to capture audio, video, screen activity, and keyboard/mouse input while completing the instrument.

The moderator left the instrument ready for completion on the home page, signaled the moment the participants should start filling out the instrument, recorded the time spent filling out the instrument, and clarified any doubts the participant had. The two observers took notes on which items and aspects of the instrument the respondents found most easy and difficult in responding to.

Four domains were evaluated regarding the user interface in accordance with what was proposed by Nielsen [78], which are learning, efficiency, errors, and satisfaction. All pertinent observations regarding completion were recorded on a specific form. Observations such as navigation difficulties, elements such as colors, questions, figures, and icons that caused doubts, and other perceived difficulties were also recorder. In addition to this, the observers scored according to the four-level performance scale, with a level proportional to the ease in answering the instrument’s question, as proposed by Da Costa et al. [67]. Technical problems, such as internet signal failures or misconfiguration of the tool on the computer, were also advised to be recorded by the observers. After answering the instrument’s questions, each participant received a brief form containing questions about their perception of recording responses on the instrument. At the end of the test, all data was tabulated in a Microsoft Excel^®^ (v. 2021) spreadsheet.

### 2.7. Ethical Aspects

This research followed all ethical precepts [79] and was approved by the Human Research Ethics Committee of Federal University of Santa Catarina (approval registration number: 6,254,504), São José Municipal Public Hospital (approval registration number: 6,292,993), and Oncology Research Center (approval registration number: 6,546,440) and received also consent to carry out the research by the Hematology and Oncology Center (project considered by the ethics committee of São José Municipal Public Hospital, 6,293,000). Only individuals who signed the Informed Consent Form of each specific phase participated in the current study’s data collection.

### 2.8. Data Analysis

For content validity analysis, the CVI tool was applied to items (CVI-items) and scales (CVI-scale). The CVI-items and CVI-scale were calculated before and after the experts’ meeting, and the acceptable values for these indices were set as from 0.78 to 1.0 [34,65,80,81]. After calculating the CVI-items, the Universal Agreement and CVI-scale were calculated. In case of inadequacy of content, the items were reformulated and went through a new round of evaluations.

In the pilot study, the usability and acceptability questions were based on the Portuguese version of the System Usability Scale (SUS) [82], a tool originally developed by Brooke [83] and widely used in usability studies of health-related applications and systems [84,85]. The instrument’s usability was assessed by calculating the total SUS score and the score for each usability attribute. The usability data recorded by the observers, such as learning, elements that caused doubts, efficiency, types of errors made, and perceived satisfaction, were described in a specific table as the key observations of the participants. For the convergent validity, previous research has demonstrated that self-reported anthropometrics can be valid enough compared to measured anthropometric markers [86,87]. In this way, associations were made between the instruments’ self-reported anthropometric measures and body image perception and the measured anthropometric data using Pearson correlations and linear regression models (crude and adjusted for age and average monthly family income). When applicable, the Bland and Altman [88] graphs were also applied and observed. A descriptive table of the pilot study was prepared, and for the usability and acceptability tests, the sample characterization data were presented according to the SUS score tertiles; for the convergent validity test, the total score of the developed instrument (0 to 10 points) was divided into tertiles to present the descriptive data of the sample at this stage. To analyze the lifestyle habits of the participants in the pilot study (initial usability testing sample), the data obtained from the developed instrument were also described in relative and absolute frequencies.

The analysis of the final usability test involved the same techniques adopted in the pilot usability study, as well as some additional analyses. The performance scale score considered a maximum individual score per screen of 3 points, with the maximum total score being 90 points (score proportional to the participant’s agility and independence in filling out the instrument) [64], since the instrument had 30 different screens. The performance scale score was described by subgroups of the following variables: number of people living in the same house (categorized as 1 to 2; ≥3); total computer usage time, categorized as 0 to 1, >1 to < 3, and ≥ 3 h/day; total cell phone usage time (0 to 1, >1 to < 3, and ≥ 3 h/day); average monthly family income up to <5 minimum wages and up to ≥5 minimum wages; total filling time in minutes (0 to 10; ≥11 and <20; ≥20); total number of clicks (0 to 90; ≥91); text inputs (0 to 40; ≥41); and frequency of assistance during filling (0 to 15; ≥16 times). The total computer usage time (hours/day), as well as the total cell phone usage time (hours/day), were calculated as follows: weekly frequency of equipment use × number of daily hours of use/7.

In all statistical analyses, the significance level considered was *p* < 0.05, and analyses were performed using the Stata^®^ statistical program (v.14.0).

## 3. Results

### 3.1. Content Validity

The content validity expert committee included seven professionals, consisting of five nutritionists, one nurse, and one psychologist. All specialists have experience in the oncology sector. After consensus meetings, the name “PrevCancer” was defined for the instrument, referring to breast cancer prevention. The main points highlighted by the experts included the need for more examples of food portions, such as fruits and vegetables, in homemade measurements. It was suggested to select examples of foods usually consumed by the Brazilian population, such as those included in the Food Guide for the Brazilian Population [49,50]. Regarding physical activity, it was advised to include more examples of moderate and vigorous physical activity, and to this end, the research team added examples of physical exercises according to AICR [48]. In questions about food consumption, it was suggested to use popular terms such as “box juice” and “fast food” to facilitate the participants’ understanding. The initial version of the questionnaire before the consensus meetings with the expert committee resulted in a CVI-items of 0.98 and a CVI-scale of 0.88, representing acceptable indices of content validity.

Reformulations to the questionnaire were performed considering the experts feedback, and the adjusted instrument was presented at a meeting with the committee. The final CVI-items and CVI-scale values resulted in 1.0, which represents an excellent content validity index. The participants’ feedback was also validated by experts in relation to the content. No aspects were highlighted to be reformulated by the experts regarding the feedback content.

### 3.2. Pilot Study

The instrument was tested for initial usability by 65 women with a mean age of 31 years (19 to 69 years old). The characterization of the sample is shown in Appendix A. The SUS score demonstrated a minimum value of 22.5 points and a maximum of 100 points, with a mean score of 88.1 (standard deviation: 17.9). Appendix A shows the usability analyses of the pilot study regarding the distribution of responses and main qualitative observations indicated in the acceptability test by the participants. It was found that most participants loved the appearance of the instrument (n = 35, 56%), had no difficulties filling it out (n = 57, 88%), and had a good understanding of the questions (n = 37, 60%). No technical problems were noticed by most participants (n = 60, 92%), and among those who experienced technical difficulties, the reason was an Internet failure. Most instrument users in the pilot study felt calm during the response period (n = 61, 94%), though some participants (n = 4, 6%) reported agitation due to fear of the result of the questionnaire. The general experience of using the instrument was great for most participants, who indicated that they loved using the tool (n = 34, 54%). Regarding this last item, some users mentioned that they found the instrument a little extensive. Other participants brought suggestions for improvements regarding the inclusion of an answer option for vegetarians/vegans in the question about red meat consumption and highlighted the importance of disseminating the instrument to more people, considering the quality of the tool (Appendix A).

The lifestyle data obtained in the pilot test with the convenience sample indicated that the total PrevCancer score had an average of 6.3 points (standard deviation: 1.1). Regarding nutritional status, most of the sample was in the healthy body weight range according to BMI (n = 40, 60.6%), with body images 1–5 on the Stunkard scale being indicated by most participants (n = 55, 83.3%). Consumption of non-starchy fruits and vegetables was reported as ≥400 g/day by 51.5% of women (n = 34). Most participants also reported eating beans five times a week or more (n = 38, 57.5%), sugary drinks once a week or less (n = 43, 65.1%), and processed meat rarely or never (n = 37, 56%). The consumption of alcoholic beverages was reported by 53% of the sample (n = 35), and the frequency of chimarrão consumption was reported as rarely or never (n = 53, 80.3%).

The convergent validity stage of the pilot study included 55 participants with a mean age of 31 years (19 to 69 years old) (Appendix A). Pearson’s correlations (Table 1) demonstrated a positive association between the PrevCancer parameters of body weight, BMI, and SSS when these were compared with the reference methods of body weight, BMI, and waist circumference obtained by anthropometric assessment (*p* < 0.001).

Table 2 shows associations through linear regression models between PrevCancer instrument parameters and reference methods. We found a significant association for all parameters (*p* < 0.001) in crude and adjusted models.

Two Bland and Altman plots were developed and are shown in Figure 2. The first graph shows the representation between measured and automatically calculated BMI in the PrevCancer instrument using self-reported data, with the average difference between the BMIs being −0.27 Kg/m^2^, the lower limit of agreement being −2.84, and the upper limit of agreement being 2.29 (Figure 2). Most BMI values were close to the difference between the means and within the range of agreement limits, indicating good agreement between the methods. Regarding the relationship between measured and reported body weight in the PrevCancer instrument (Figure 2), the average difference in body weight was −0.21 Kg, the lower limit of agreement was −4.94, and the upper limit of agreement was 4.51, with most values concentrated close to the line of average values and within the range of agreement limits (Figure 2). A divergent pattern was noticed in both Bland and Altman plots, suggesting that the differences between the two instruments became larger with higher BMI and weight values.

### 3.3. Instrument Adaptation

Pilot usability and acceptability studies provided adjustment directions for some specific issues of the PrevCancer instrument. For the breastfeeding question, if the participant does not have children, a specific response field was inserted to cover this situation. Based on participants’ reports of preferring to know how many questions the instrument has and how long it takes to answer them, a sentence was included on the questionnaire’s home page mentioning the number of questions and average response time. The participants’ recommendations also reinforced the need to address issues about physical activity more clearly, such as calculating the time spent exercising in a week. In this sense, more examples of how to calculate the total time in minutes of physical activity practiced during the week were included in the questions.

The second digital structuring gave rise to the PrevCancer instrument in the format of a website (Figure 3). We decided to have the pink bow as the symbol of the instrument in reference to cancer prevention. All questions were designed in larger fonts for easy viewing by participants, and images from the Canva graphic design platform were included to help participants understand the questions.

### 3.4. Final Usability Study

The final usability stage involved 14 breast cancer patients (average age 54 years, 40 to 73 years old). Among these women, nine patients (64%) were from public health institutions. The average SUS score was 90.3 points (65 to 100 points, standard deviation = 2.5). Table 3 shows the characterization of the sample according to the SUS score tertiles.

We found that most patients loved using the PrevCancer tool (n = 8, 57%), showed no difficulties completing it (n = 9, 64%), had no technical difficulties with the instrument (n = 12, 86%), and felt calm during the questionnaire response period (n = 11, 79%) (Table 4).

It was noticed that one of the most common difficulties for the participants was not seeing the observations of the questions that provided additional explanations to enable the answer, which may have occurred due to the font size that must be increased, according to the participants’ notes. The participants’ receptivity to the instrument was considered excellent, as everyone (n = 14, 100%) responded that they liked or loved the general experience of using the instrument. Some of the participants’ feedback included the identification of lifestyle habits to be improved based on the final grade for each component evaluated and interest in filling out the tool again in the future to compare results (Table 4).

The average performance scale score was 71 points (0 to 90 points, standard deviation = 23.8). One of the participants was unable to fill out the instrument alone due to difficulties with electronic equipment, despite claiming ability to use the computer. The average time to complete the instrument was 17 min (8 min to 30 min, standard deviation = 1.6). The data in Table 5 show higher performance scale score values for those who live with fewer people in the same house, with higher total computer and cell phone usage time, and higher average monthly family income. Participants who completed the instrument in less time, with fewer text inputs, and with less dependence on assistance had a better score on the performance scale recorded by observers (Table 5).

The participants’ most common difficulties with the questionnaire were in relation to placing height information in centimeters instead of meters (n = 10; 71%), classifying the intensity of the physical activity practiced (n = 7; 50%), and not filling in the answer “zero” in case of no consumption of one or more types of alcoholic beverages (n = 9; 64%). Furthermore, a few participants had difficulties in converting hours of moderate physical activity into minutes (n = 5, 36%), and doubts arose about which foods are involved in the question about frequency of consumption of beans (n = 6, 43%). It took longer to answer questions related to the city where they live and their profession, since the instrument has fixed lists of response options which, in the participants’ opinion, are very extensive (n = 5, 36%).

Finally, the participants suggested the inclusion of an answer option “I don’t consume sugary drinks”; the exchange of the term “beans” for “grains” and providing examples of foods from this group in the question statement; and larger font in the observations that were contained on the question pages.

The results obtained through PrevCancer with the final usability test sample showed an average score of 7.1 points (standard deviation: 0.8). Most patients were overweight (n = 10, 71.4%), even though body images of healthy weight were frequently indicated on the Stunkard scale (healthy body weight images 1 to 5, n = 9, 64.3%). Most patients reported daily consumption of non-starchy fruits and vegetables ≥ 400 g/day (n = 11, 78.6%). Half of the sample (n = 7, 50%) reported consuming beans five times or more per week. The frequency of consumption of sugary drinks was reported by most participants as once a week or less (n = 10, 71.4%). It was found that the answer “rarely or never” was attributed by most women to the consumption of processed meats (n = 9, 64.2%) and chimarrão (n = 10, 71.4%). The consumption of alcoholic beverages was confirmed by most participants (n = 10, 71.4%).

## 4. Discussion

In our understanding, the present study gave rise to the first digital instrument to evaluate the lifestyle habits of Brazilian female breast cancer survivors based on WCRF/AICR recommendations. Relevant aspects of diet and nutritional status, such as body weight, consumption of non-starchy fruits and vegetables, and consumption of sugary drinks, chimarrão, and ultra-processed foods were operationalized in the instrument through objective questions to participants. The self-assessment PrevCancer digital instrument had an acceptable content validity as assessed by a committee of experts, presented good usability and acceptability for the general public and breast cancer patients, and showed evidence of convergent validity for the anthropometric data. Performance in filling out the instrument was greater among those with higher average monthly family income and with higher total computer and cell phone usage time, that is, those who use computers and cell phones more often. These participants answered the instrument in less than 20 min, had fewer text entries, and required less assistance while completing the instrument.

We recognize that the results of the lifestyle assessment obtained in PrevCancer can guide specific lifestyle interventions according to the identification of areas for improvement. This systematic health data generated by PrevCancer meets the definition of precision health, which comprises an emerging field aiming to promote the well-being of the population through the monitoring of health data to design more effective treatments [89].

The average PrevCancer scores of the pilot study and the final usability test samples indicated a moderately healthy lifestyle. In this context, a prevalent aspect among the participants was the weekly consumption of alcohol, which is not recommended by the WCRF/AICR [7]. Consumption of alcoholic beverages has a negative impact on the DNA repair mechanism and increases estradiol levels and the risk of some diseases, such as liver disease and pancreatitis [18]. Although the literature suggests that low doses of alcohol may be beneficial for some health outcomes [90], such as serum blood glucose levels and risk of acute myocardial infarction, it is known that no dose of alcohol is safe regarding cancer prevention [7]. The WCRF/AICR recommendations report from a Brazilian perspective [18] reinforces that any type of alcoholic beverage has a similar impact on the risk of cancer. Regarding the nutritional status of the participants, it is noteworthy that the final usability test sample showed a prevalence of overweight. This is in line with previous studies showing that more than 50% of breast cancer patients gain significant weight after the diagnosis of the disease [91], which is largely influenced by the medication regimens to which they are subjected [92]. Excess weight can influence patients’ worse survival, which highlights the need for effective nutritional strategies to control these women’s body weight [93].

However, it is noteworthy that some protective factors against cancer recurrence were highly prevalent in the samples of this study. Most participants reported adequate consumption of legumes (five times or more per week), meeting the recommendation from the Food Guide for the Brazilian Population [49,50]. Furthermore, the consumption of fruits and vegetables exceeded 400 g/day by most participants, which differs greatly from the findings in the literature. The study by Gali et al. [94], who evaluated adherence to the WCRF/AICR recommendations in women with breast cancer and without the disease, showed that, among those who had the diagnosis, the recommended consumption of fruits and vegetables was only achieved by 19.9% of the sample. In women who did not have breast cancer, the consumption of non-starchy plant foods was even lower (18%) [94]. In contrast, low consumption of sugary drinks was prevalent [94], exceeding 67% of the women in the study with and without breast cancer, similar to the present study.

When comparing the PrevCancer instrument with other tools already published, we emphasize that our digital solution presents particularities that significantly differentiate it from others. At a global level, the existence of two instruments designed to assess lifestyle in relation to preventing cancer/recurrence was identified: the “Cancer Health Check” [22] developed in England and the “Prévention cancers: le test” [23] prepared in France. However, these instruments do not include specific recommendations adapted to the Brazilian reality, such as the consumption of chimarrão, very common in the southern region of Brazil, and the consumption of legumes, considering that this food group typically makes up Brazilian dishes [18,50]. Another instrument worth mentioning is The Energy Balance on Cancer mobile app, aimed at breast cancer survivors [26,27,28]. The application’s variables emphasize food consumption and physical activity to analyze the patient’s energy balance, based on the recommendations of the American Cancer Society [95] and WCRF [7,96]. It is important to highlight the significance of tools in this regard that provide new forms of support for breast cancer survivors. In this context, we emphasize the relevance of the PrevCancer instrument to address the Brazilian context through the assessment of specific lifestyle habits, especially of women in the southern region of the country.

During the development of the tool, we prioritized a “person-based” approach, evaluating the psychosocial context of women through their qualitative assessments of the questions of the instrument [97]. In patients with breast cancer, digital self-assessment has already been recognized as an opportunity to reflect on themselves and understand their own health status, as pointed out by Kanakubo et al. [98] in a descriptive study involving self-assessment using a digital questionnaire for Japanese women with breast cancer between 20 and 75 years old. These insights allow the design of the new tool presented in this study, which is understandable and acceptable by the target audience. Indeed, it was found that the PrevCancer instrument was well accepted by the participants (general public and breast cancer participants) when analyzing the average SUS values and the percentages of responses using the Likert scale for each evaluated usability attribute. The SUS averages found in the current study were higher than what was identified in other studies [99,100] on the usability of self-assessment health tools, which considered values around >70 acceptable. Similar to the findings of Signorelli et al. [99] in a usability evaluation of a mobile intervention focused on physical activity for breast cancer patients, the participants in the present study reported that they considered the instrument useful and user-friendly.

Among the main observations qualitatively indicated by the participants, in both the pilot and final usability studies, there was the need to adapt the instrument to questions involving numerical calculations, such as the sum of weekly minutes of physical activity, for example, and an expansion of the response possibilities for questions relating to red meat and alcoholic drinks consumption. The difficulty in reporting the level of physical activity practiced has also been highlighted in the study by Huijnen et al. [101] with patients with chronic pain. In the context of Brazilian adults, Matsudo et al. [102] state that most individuals have difficulty in differentiating moderate from vigorous physical activity, as well as in quantifying the period of time practiced, especially when they are shorter periods of time (less than 10 min daily), which leads, in many cases, to overestimation of physical exercise. As for answer options, Kachouei et al. [103] also faced the need to include more possibilities to respond to what participants demand in a mobile lifestyle assessment app for breast cancer survivors. Regarding the difficulties encountered in answering questions about physical activity, the importance of adapting the answers to more intuitive options is highlighted. This holds significant importance as users are more prone to advocate for products to others predicated on intuitive affective responses. Elevated perceived usability has the potential to augment customer allegiance [104,105].

Regarding the participants’ performance when filling out the instrument in the final usability test, the average performance score was similar to that found by Da Costa et al. [67], who also used this scoring method when applying an online questionnaire to an audience of schoolchildren (average score of 73 points). In the current study, participants with longer total time of daily computer and cell phone use, lower total response time, and less dependence on assistance in answering performed better in the PrevCancer instrument. Moon et al. [106] highlights that online surveys involving breast cancer survivors should always consider factors such as access to digital devices and affinity with technology so that greater involvement of respondents is promoted. In this sense, minimizing technological barriers and increasing digital literacy can be effective ways to improve care in relation to cancer treatment [107,108].

When analyzing the convergent validity results, the isolated parameters of body weight, BMI, and SSS measured by PrevCancer demonstrated good correlation with the reference methods obtained by anthropometric assessment. The pattern found in the graphical representation using Bland and Altman analysis suggests substantial agreement between the self-reported and measured anthropometric data, indicating that the PrevCancer anthropometric measures are valid. In a validity study involving American adults, Hodge et al. [109] demonstrated that self-reported body parameters demonstrate validity as measures in both men and women across diverse socio-demographic scenarios. Other studies [110,111] also concluded that self-reported weight and height data were valid for demonstrating nutritional status in epidemiological studies. These findings corroborate the feasibility of using self-reported body weight measurement in different study contexts. Regarding the relationship between measured waist circumference and reported body image data, Sutcliffe et al. [112] have already demonstrated a good correlation between self-reported body size, based on the SSS, and waist circumference. Other previously cited studies also demonstrated this agreement [45,46,47], making the body image question a useful alternative to waist circumference data in the digital instrument.

We highlight that this is the first instrument to provide an assessment of adherence to the WCRF/AICR [7] recommendations in the Brazilian context of breast cancer survivors, which demonstrated good usability, acceptability and convergent validity. West and Michie [113] state that as soon as the first components of the digital instrument are established, they must begin to be tested and refined repeatedly. In this sense, this study compiles the first stages of the development the PrevCancer instrument, with methodological steps that allowed the improvement and refinement of the instrument. The development of this instrument is pertinent because although it considers global recommendations for preventing cancer and recurrence [7], it also considers national healthy eating guidelines from the Food Guide for the Brazilian Population [49,50], which makes the lifestyle assessment more appropriate to the lifestyle of Brazilian women diagnosed with breast cancer.

We recognize that our study has some limitations. Considering the convergent validity in the pilot study, initial tests already show promising directions regarding the tool’s validity. However, more reference methods must be adopted to cover the different domains of PrevCancer, regarding food consumption, physical activity, smoking and drinking habits, and sun exposure, for example. It should be noted that other reference methods were not applied for now as the instrument was tested in an academic week event which was open to the community, with brief approaches to participants at a specific stand of the research group. Another important aspect concerns the biases of the self-reported data in PrevCancer, which corroborates the need for new validation studies using different reference methods regarding the assessment of lifestyle habits. We also consider the relatively small sample size for final usability testing, which, despite the intended sample size being based on the literature, could be further expanded. However, we identified the main points of improvement in the instrument through usability testing, and we also tested the instrument on a larger female population sample through pilot testing. New studies involving PrevCancer must be designed to advance validation analyses and to evaluate the reproducibility of the tool.

## 5. Conclusions

The PrevCancer instrument was shown to have valid content and a great usability and acceptability for the target population, proving to be a useful tool for future research involving the lifestyle of breast cancer survivors. Completing the instrument and knowing the results regarding following the WCRF/AICR [7] recommendations may provide clarification regarding the lifestyle panorama and may contribute to health promotion among breast cancer survivors based on the interpretation of the results at the individual level. At a population level, the proposed assessment instrument can be an important tool for both cancer-related studies, such as cohort studies to monitor the lifestyle of breast cancer survivors, and cross-sectional observational studies on the assessment of behavioral factors in specific health services, such as interdisciplinary clinics and hospitals. The results obtained by systematized health assessment can provide support for the implementation of nutritional, medical, and physical activity interventions and thus strengthen cancer prevention in different scenarios. Future study perspectives on this instrument include reproducibility and validity tests covering all components of the instrument.

## 6. Patents

A request was made to deposit a computer program (PrevCancer) at the National Institute of Industrial Property of Brazil (patent application process protocol number: 512024002453-0).

## Figures and Tables

**Figure 1 nutrients-16-03707-f001:**
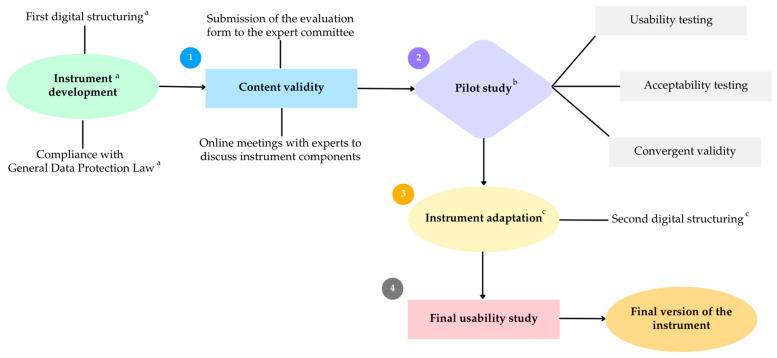
Diagram with the phases of the study. ^a^ All items related to the instrument development consisted of methodological stages of the study. An initial version of the instrument was structured on the Google Forms platform to be subjected to pilot study. ^b^ Pilot study conducted with an initial version of the instrument structured in Google Forms. ^c^ The partial version of the instrument was replaced by the official platform version from step 3 onwards.

**Figure 2 nutrients-16-03707-f002:**
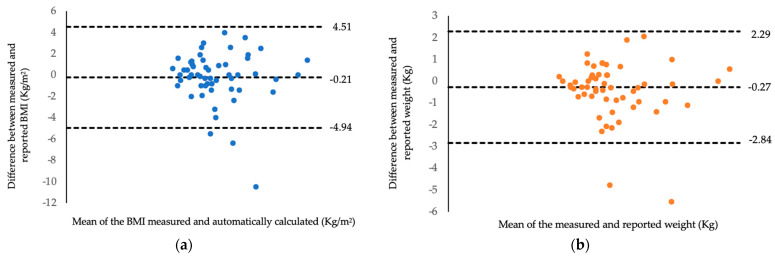
Bland and Altman graphs on the relationship between the reference method and PrevCancer instrument parameter. (**a**) Relationship between measured and automatically calculated BMI in the PrevCancer instrument. (**b**) Relationship between measured and automatically calculated body weight in the PrevCancer instrument.

**Figure 3 nutrients-16-03707-f003:**
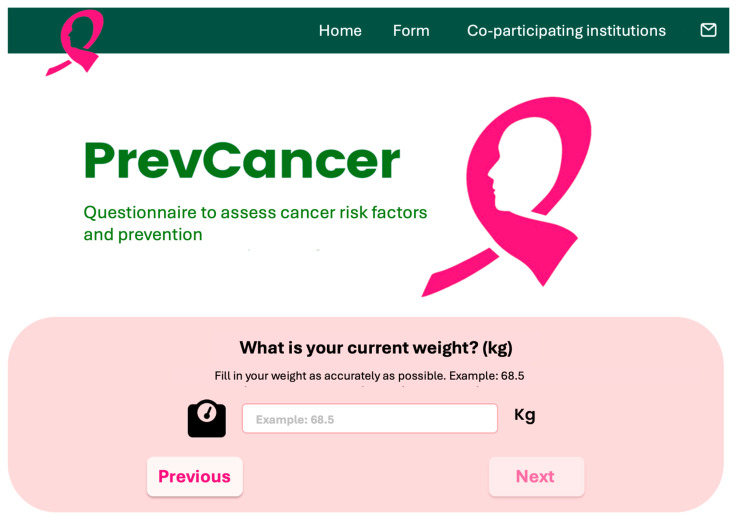
PrevCancer website layout example (adapted to English language), a self-assessment tool for the lifestyle assessment of breast cancer survivors.

**Table 1 nutrients-16-03707-t001:** Pearson’s correlations coefficients between PrevCancer instrument parameters and reference methods obtained by anthropometric measurement assessment (n = 55).

Variables	PrevCancer Body Weight	PrevCancer BMI	PrevCancer SSS	Measured Body Weight	Measured BMI	Measured Waist Circumference
PrevCancer body weight	1	0.93 *	0.78 *	0.98 *	0.92 *	0.86 *
PrevCancer BMI ^1^		1	0.83 *	0.91 *	0.97 *	0.89 *
PrevCancer SSS ^2^			1	0.81 *	0.85 *	0.82 *
Measured body weight				1	0.94 *	0.88 *
Measured BMI ^1^					1	
Measured waist circumference						1

^1^ BMI = body mass index. ^2^ SSS = Stunkard Silhouette Scale. * *p* < 0.001. All analyses were performed using Pearson correlation.

**Table 2 nutrients-16-03707-t002:** Associations between PrevCancer parameters and reference methods for convergent validity.

PrevCancer Parameters		Reference Method ^a^								
		Body Weight ^a^			BMI ^a^			Waist Circumference ^a^		
	Model	β	CI 95% ^2^	*p* ^1^	β	CI 95%	*p* ^1^	β	CI 95%	*p* ^1^
**Body Weight**	Basic	0.96	0.92–1.01	**<0.001**	2.18	1.94–2.42	**<0.001**	0.89	0.74–1.03	**<0.001**
	Adjusted	0.96	0.91–1.01	**<0.001**	2.17	1.92–2.42	**<0.001**	0.92	0.77–1.07	**<0.001**
**BMI** ^1^	Basic	0.37	0.32–0.41	**<0.001**	0.94	0.88–1.01	**<0.001**	0.38	0.32–0.43	**<0.001**
	Adjusted	0.37	0.32–0.41	**<0.001**	0.94	0.87–1.00	**<0.001**	0.39	0.33–0.44	**<0.001**
**SSS** ^3^	Basic	0.09	0.07–0.1	**<0.001**	0.22	0.19–0.26	**<0.001**	0.09	0.07–0.11	**<0.001**
	Adjusted	0.08	0.07–0.1	**<0.001**	0.22	0.28–0.26	**<0.001**	0.09	0.07.0.11	**<0.001**

^1^ BMI = body mass index. ^2^ CI 95% = 95% confidence interval. ^3^ SSS = Stunkard Silhouette Scale. ^a^ Measured parameters obtained by anthropometric assessment. *p*^1^ refers to the associations of unadjusted linear regression models. *p* refers to the associations of linear regression models (crude and adjusted for age and average monthly family income). Values with *p* < 0.05 were considered significant and are highlighted in bold.

**Table 3 nutrients-16-03707-t003:** Characteristics of participants in the final usability study of a self-assessment instrument for the lifestyle of breast cancer survivors (PrevCancer) according to the System Usability Scale score tertiles.

Variables	Total (N = 14)	System Usability Scale Score		
		Tertile 1 (Score 65 to 92.5, n = 8)	Tertile 2 (Score 95, n = 2)	Tertile 3 (Score 97.5 to 100, n = 4)
**Age (years)** ^a^	-	53.6 (4.1)	55 (1.0)	55.7 (1.7)
**Education** ^b^				
Incomplete 2nd degree	2 (100)	1 (50)	0 (0)	1 (50)
Complete 2nd degree	4 (100)	2 (50)	1 (25)	1 (25)
Complete graduation	8 (100)	5 (62.5)	1 (13)	2 (25)
**Race/ethnicity** ^b^				
White, non-Hispanic	7 (100)	5 (71)	0 (0)	2 (29)
Black, non-Hispanic	1 (100)	0 (0)	1 (100)	0 (0)
Other, non-Hispanic	6 (100)	3 (50)	1 (17)	2 (33)
**Marital status** ^b^				
Married/stable union	5 (100)	3 (60)	0 (0)	2 (40)
Not married/no stable union	9 (100)	5 (56)	2 (22)	2 (22)
**Average monthly family income** ^b^				
Up to <5 minimum wages	6 (100)	3 (50)	2 (33)	1 (17)
Up to ≥5 minimum wages	8 (100)	5 (62)	0 (0)	3 (38)
**Menopause status** ^b^				
Pre-menopause	4 (100)	1 (25)	1 (25)	2 (50)
Post-menopause	10 (100)	7 (70)	1 (10)	2 (20)
**Staging of primary breast cancer** ^b^				
I	10 (100)	7 (70)	1 (10)	2 (20)
II	4 (100)	1 (25)	1 (25)	2 (50)
III		-	-	-
**Lymph node involvement** ^b^				
Yes	4 (100)	2 (50)	1 (25)	1 (25)
No	10 (100)	6 (60)	1 (10)	3 (30)
**Clinical subtype of breast cancer** ^b^				
Luminal A	10 (100)	5 (50)	2 (20)	3 (30)
Luminal B	2 (100)	2 (100)	-	-
HER2+	-	-	-	-
Triple-negative	2 (100)	1 (50)	0 (0)	1 (50)
**Number of radiotherapy sessions** ^a^	-	19.6 (2.9)	16.5 (8.5)	21.6 (4.6)
**Number of chemotherapy sessions** ^a^	-	13.7 (1.6)	20.5 (4.5)	13.5 (3.7)
**Performance scale score** ^a^	-	67.7 (10.6)	73.5 (12.5)	79 (5.9)
**Total filling time (minutes)** ^a^	-	19.2 (2.3)	18.5 (1.5)	13.2 (2.5)
**Total number of clicks** ^a^	-	90.2 (13.8)	124.5 (55.5)	80.7 (5.4)
**Text inputs** ^a^	-	41.2 (3.2)	47 (15)	38 (3.3)

^a^ Results presented in mean and standard deviation. ^b^ Results presented in n and %.

**Table 4 nutrients-16-03707-t004:** Usability analysis of a self-assessment instrument for the lifestyle of breast cancer survivors (PrevCancer).

Usability Attribute	n (%)	Key Observations from the Participants
**Appearance of the tool**		“The instrument is very clear and organized.” “I was unable to clearly visualize the additional information on the questions that were in green font.” “I didn’t think the image of the beans was very clear.”
1—I hated it	-	
2—I did not like it	-	
3—Indifferent	1 (7)	
4—I liked it	5 (36)	
5—I loved it	8 (57)	
**Difficulty in filling out the instrument**		“I had difficulty classifying the intensity of physical activity performed.” “I was unable to locate additional observations to assist me in answering.”
No	9 (64)	
Yes	5 (36)	
**Understanding of the questions**		“I was unsure about which foods were all being considered in the question about beans.”
1—I hated it	-	
2—I did not like it	-	
3—Indifferent	-	
4—I liked it	8 (57)	
5—I loved it	6 (43)	
**Technical problems with the instrument**		
No	12 (86)	“The final feedback report on question results did not download automatically.”
Yes	2 (14)	
**Feeling when filling out the instrument**		
Calm	11 (79)	“I was a little nervous because my eye was a little irritated on the day of the interview.”
Agitated	2 (14)	
Other	1 (7)	
**General experience of using the instrument**		“I found it very easy to fill out the questionnaire, as well as being very enlightening!” “I was happy with the result of the self-assessment and as a result it helped me better observe my lifestyle habits.” “Great to know that I need to improve some parameter, in this case, the consumption of beans.” “I enjoyed using the instrument and was interested in filling it out again in the future to compare results.
1—I hated it	-	
2—I did not like it	-	
3—Indifferent	-	
4—I liked it	7 (50)	
5—I loved it	7 (50)	

**Table 5 nutrients-16-03707-t005:** Performance scale score by characteristics of participants in a study involving the self-assessment instrument for the lifestyle of breast cancer survivors (PrevCancer).

Variable	N (%)	Performance Scale Score		
		Mean (Standard Deviation)	Min	Max
**Number of people living in the same house**				
**1 to 2**	10 (71)	76 (13.5)	50	90
≥3	4 (29)	61.2 (41.1)	0	88
**Total computer usage time (hours/day)**				
0 to 1	7 (50)	63.2 (29.2)	0	85
>1 to <3	2 (14)	62.5 (17.6)	50	75
≥3	5 (36)	87.4 (2.4)	84	90
**Total cell phone usage time (hours/day)**				
0 to 1	4 (29)	46.7 (32.9)	0	76
>1 to <3	3 (21)	75.6 (12.1)	64	88
≥3	7 (50)	84.4 (4.9)	75	90
**Average monthly family income**				
Up to <5 minimum wages	6 (43)	61.5 (31.7)	0	86
Up to ≥5 minimum wages	8 (57)	79.5 (13.3)	50	90
**Total filling time (minutes)**				
0 to 10	3 (21)	89 (0.6)	88	90
≥11 <20	5 (36)	78.8 (4.9)	75	86
≥20	6 (43)	57.3 (31.2)	0	85
**Total number of clicks**				
0 to 90	10 (71)	70.6 (26.6)	0	89
≥91	4 (29)	74.7 (17.6)	50	90
**Text inputs**				
0 to 40	7 (50)	77.2 (10.9)	61	88
≥41	7 (50)	66.3 (32.2)	0	90
**Frequency of assistance during filling**				
0 to 15	12 (86)	78.6 (11.8)	50	90
≥16	2 (14)	30.5 (43.1)	0	61

## Data Availability

The original contributions presented in the study are included in the article and Appendix A, further inquiries can be directed to the corresponding author.

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
