# Peer review of "Development, Content Validity and Usability of a Self-Assessment Instrument for the Lifestyle of Breast Cancer Survivors in Brazil"

_nutrients, 2024, doi:10.3390/nu16213707_

Round 1
Reviewer 1 Report
Comments and Suggestions for Authors
Patricia Faria Di Pietro submitted to Nutrients a manuscript, dealing with a self-assessment instrument for the lifestyle of breast cancer survivors.
The paper is written in a robust and detailed manner, the investigation appears to be carried out and described correctly, providing added value to experts in this field.
The title should indicate that the study refers to a Brazilian territorial context; furthermore, I ask to evaluate the opportunity to increase the bibliography with references published in print in 2024 (not only sitography).
Comments on the Quality of English LanguageMinor editing of English language required.
Reviewer 2 Report
Comments and Suggestions for Authors
According to the journal’s guidelines, the abstract word limit is 200. Please revise it.
You need to provide more keywords to identify your study.
Once you submit your work to Nutrients, a journal focusing on human nutrition, you should align your manuscript with this. There isn’t a single mention of nutrition in your Introduction. The background should also be enriched to give the needed justifications to carry out your investigations.
The applied methodologies seem adequate and are properly described.
It is not possible to understand Figure 3. Can you please improve its quality and size?
The Results are pertinent. However, more discussion is necessary from a nutritional perspective.
Despite your work's relevance, Nutrients journal is not the best way for you to publish it. The main focus of your study is not aligned with the main focus of Nutrients.
Reviewer 3 Report
Comments and Suggestions for Authors
Thank you for the opportunity. There are several areas where the manuscript needs further improvements.
---------------
1- The introduction provides adequate background on the prevalence of breast cancer and the role of lifestyle in the prevention of recurrence. However, the rationale for selecting specific items from the WCRF/AICR recommendations—in particular, the Brazilian adaptations—needs to be made a bit clearer. For instance, the inclusion of items such as 'chimarrão consumption' requires more explanation in terms of local relevance and prevalence.
2- The process for the content validity needs additional explanation. It states that the calculations of CVI are performed, but further detail on the selection of the expert panel and how comments were integrated into these calculations should be given. Were there any disagreements that arose between experts, and how were they resolved? Also, the demographic breakdown in the pilot study is very detailed, but a fuller justification of sample size for usability—n = 65—stronger the methodology. Was this based on a power calculation or prior research? This aspect isn't clearly addressed.
3- It is commendable that the paper contains psychometric analyses for the validity of the 'PrevCancer' tool, but some explanation of convergent validity tests conducted should be provided. How does one select anthropometric measures as a valid comparator? Were there other tests beyond correlation, such as regression analysis?
4- The discussion has contextualized the 'PrevCancer' tool within a wider landscape of cancer self-assessment tools, but more could be done to robustly discuss what this tool might mean for clinical and public health practices. How might it be used to improve long-term health outcomes or inform clinical decisions?
5- There is a lack of discussion on the study's limitations, which could have been elaborated on in greater detail. To be precise, the limitations section needs more critical discussion regarding self-reported data, such as bias or inaccuracies, and a relatively small sample size for final usability testing (n=14).
Round 2
Reviewer 2 Report
Comments and Suggestions for Authors
The authors have considerably improved the manuscript, focusing it from a nutritional perspective, so I am satisfied with the revisions made.
Author Response
Dear Reviewer,
We greatly appreciate your comments and contributions to our paper. We are pleased to know that our manuscript has improved following your suggestions, and we have addressed all your recommendations carefully.
Thank you once again for recognizing the value of our study.
Yours sincerely,
The Authors
Reviewer 3 Report
Comments and Suggestions for Authors
Thank you for all the edits and answering in detail to my suggestions.
Author Response

(The authors gave the same response as above.)
